# Distinguishing gene flow between malaria parasite populations

**Tyler S. Brown**[1,2]*, **Olufunmilayo Arogbokun**[3], **Caroline O. Buckee**[1], **Hsiao-Han Chang**[1,4]*

**1** Center for Communicable Disease Dynamics, Harvard T.H. Chan School of Public Health, Boston, Massachusetts, United States of America, **2** Infectious Diseases Division, Massachusetts General Hospital, Boston, Massachusetts, United States of America, **3** Infectious Disease Epidemiology and Ecology Lab, University of North Carolina School of Medicine, Chapel Hill, North Carolina, United States of America, **4** Institute of Bioinformatics and Structural Biology, National Tsing Hua University, Hsinchu City, Taiwan

* tsbrown@mgh.harvard.edu (TSB); hhchang@life.nthu.edu.tw (H-HC)

## Abstract

Measuring gene flow between malaria parasite populations in different geographic locations can provide strategic information for malaria control interventions. Multiple important questions pertaining to the design of such studies remain unanswered, limiting efforts to operationalize genomic surveillance tools for routine public health use. This report examines the use of population-level summaries of genetic divergence ($F_{ST}$) and relatedness (identity-by-descent) to distinguish levels of gene flow between malaria populations, focused on field-relevant questions about data size, sampling, and interpretability of observations from genomic surveillance studies. To do this, we use *P. falciparum* whole genome sequence data and simulated sequence data approximating malaria populations evolving under different current and historical epidemiological conditions. We employ mobile-phone associated mobility data to estimate parasite migration rates over different spatial scales and use this to inform our analysis. This analysis underscores the complementary nature of divergence- and relatedness-based metrics for distinguishing gene flow over different temporal and spatial scales and characterizes the data requirements for using these metrics in different contexts. Our results have implications for the design and implementation of malaria genomic surveillance studies.

## Author summary

Malaria is a leading infectious cause of illness and death worldwide. Understanding how malaria parasites are spread between different geographic locations can provide useful information for disease control efforts. Examples include identifying source locations for imported infections in lower-incidence "sink" locations and delineating the routes over which drug-resistant malaria strains disperse across geographic space. Genomic surveillance methods use geolocated genetic sequence data from malaria infections to estimate gene flow and connectivity between parasites populations in different locations. This approach has yielded important insights into patterns of connectivity between malaria

**Data Availability Statement:** All P. falciparum genetic variant call data from the Pfk3 data collection are available at https://www.malariagen.net/data/pf3k-5. Code for generating simulated sequence data used in the manuscript, and

estimated human mobility fluxes used in these simulations, are available at https://osf.io/5hwty/.

**Funding:** This work was supported by the United States National Institute of Health (www.nih.gov) grants T32AI007061 (TSB) and R35GM12471 (COB) and Ministry of Science and Technology in Taiwan grant MOST 110-2636-B-007-009 (HHC). The funders had no role in study design, data collection and analysis, decision to publish, or preparation of the manuscript.

**Competing interests:** The authors have declared that no competing interests exist.

populations over local, national, and global scales. However, there are multiple unresolved questions about the design and interpretation of these studies. This study evaluates how much data is needed to distinguish different levels of gene flow between parasite populations ("Are the malaria populations in locations $i$ and $j$ linked by higher or lower connectivity than those in locations $k$ and $l$?"). We examine data size requirements (including the number of genetic markers and number of individual infections analyzed) for this important, implementation-relevant task across multiple epidemiological scenarios, providing practical guidance for the design and interpretation of similar studies.

## Introduction

Measuring the extent to which malaria parasite populations are linked across different geographic locations ("connectivity") can provide important guidance for the design and implementation of malaria control interventions. Multiple approaches are available for this task, including methods that measure human host mobility (using census-derived [1] or mobile phone-associated mobility data [2, 3]) as a proxy for parasite migration and genomic surveillance strategies that directly measure parasite gene flow between locations.

Recent studies have combined these two approaches, using both human mobility data and estimated parasite gene flow to examine previously undescribed aspects of malaria spatial epidemiology [3] (for example, source-sink dynamics in endemic areas of Bangladesh [2]). Other recent work has focused on estimating the probability that a pair of genomes are identical by descent (IBD) at a particular locus [4] and using population-level summaries of IBD estimates to assess gene flow between malaria populations [5]. These relatedness-based approaches have uncovered spatial structure in malaria populations at small, local scales, where conventional methods using differentiation-based estimators (including the fixation index, $F_{ST}$) fail to identify population structure [5]. IBD-based measures capture variation due to recombination and as such may be well suited for measuring more recent gene flow between malaria populations (in which mutation rates are relatively slow, but variation due to recombination can accrue more quickly).

These studies and others evaluate for the presence or absence of spatial structure by evaluating for isolation by distance, i.e. by testing whether estimated pairwise differentiation or relatedness correlates with inter-location distance (for example, see [6, 7]). Importantly, methods based on isolation by distance may be difficult to interpret if migration is not spatially coherent, i.e. if human and/or parasite migration rates do not consistently correlate with distance, as has been observed in multiple contexts [8–10]. Another common approach used in malaria genomic surveillance studies classifies two malaria populations as somewhat connected (versus not connected) if the confidence interval of their pairwise $F_{ST}$ value does not include zero [11–14]. This binary approach may have limited value in situations where ranking weakly versus strongly connected populations is important. From a practical standpoint, important use cases for gene flow estimation include ranking different levels of connectivity between locations ("Are the malaria populations in locations $i$ and $j$ linked by higher or lower connectivity than those in locations $k$ and $l$?") and classifying location pairs by their relative levels of connectivity ("Among the malaria populations in locations $i$, $j$, $k$, and $l$, which of the six total location pairs are most highly connected?"). Isolation by distance can be considered a special case of ranking, in which the ranks of estimated gene flow are compared to the ranks of inter-location geographic distances.

The properties of various estimators of $F_{ST}$ have been studied extensively and there is a rich and longstanding literature on IBD [15]. However, there is currently only limited guidance on the size [5] and type [16] of genetic data needed for measuring gene flow between malaria parasite populations, and none specific to the questions of ranking and classification. Most studies of this kind use genetic data of widely varying sizes and types, including both whole genome sequence data and SNP 'barcode' data (typically consisting of 24- to 100-SNPs [17, 18]), limiting generalizability and comparability between studies. Importantly, population-level summaries of differentiation and relatedness capture genetic signatures resulting from multiple, interconnected processes, including recent prior and ancestral migration, changes in population size over time, and selection. Multiple studies indicate that effective sizes of malaria populations are dynamic, often over relatively short periods of time (for example, marked contraction of effective population size following the successful introduction of various malaria control interventions [19, 20]). For these reasons, it is likely that the amount or type of data needed for estimating gene flow, and reliably ranking or classifying these estimates, will vary across different epidemiological contexts.

This report examines the use of both relatedness- and differentiation-based methods for estimating gene flow between malaria populations. We numerically evaluate data size requirements for ranking and classification of these estimates, using *P. falciparum* genetic data from field isolates and a coalescent simulation informed by mobile phone-associated human mobility data. In addition, we explore how changes in population size and migration rates over time influence IBD- and $F_{ST}$-based estimates of gene flow. (Although a multitude of other methods and estimators exist for studying differentiation and relatedness between populations, we focus here on $F_{ST}$ given its common usage in malaria genomic epidemiology studies and IBD-based methods given their increasing use in similar applications.) We specifically examine populations that have undergone recent reduction in their effective population sizes or changes in relative parasite migration rates between locations, given the direct relevance of this scenario to real-world malaria populations. Findings from these analyses have important implications for the design and implementation of malaria genomic surveillance studies.

## Methods

### *P. falciparum* genomic data

We obtained *P. falciparum* whole genome variant call data, generated using the Genome Analysis Toolkit (GATK, [21]), from the MalariaGen database (Pf3k data release 5.1 [22]). We restricted our analysis to samples from the Greater Mekong Subregion (GMS), excluding more widely divergent *P. falciparum* populations from Sub-Saharan Africa and Bangladesh that are less relevant to the context of our analysis (i.e. genomic surveillance at the regional and national level), and including only individual sequences from clinical cases or survey participants. After excluding "un-callable" regions of the *P. falciparum* genome (i.e. highly repetitive or highly variable regions in which short-read based genotyping is unreliable [23]) and sites failing any GATK quality filter, we removed suspected polyclonal sequences and those with poor quality sequence data based on the proportions of heterozygous and missing SNP calls across all sites for each sample, and removed low-quality SNPs based on site-wise missingness and heterozygosity (similar to [24]). We first removed SNP sites for which > 1% of sequences had missing or heterozygous calls and then removed sequences for which > 5% SNP sites were missing or heterozygous. This filtering protocol yielded 472 individual sequences and 143,480 SNPs. Of the 143,480 SNPs, 437 had estimated minor allele frequencies > 0.35 and 5537 had estimated minor allele frequencies > 0.05 (estimated using all 472 sequences in this collection, Fig A in S1 Appendix). We conducted analyses with both of these SNP sets, with the goal of

examining how SNP allele frequency influences gene flow estimates. In terms of precision, markers with equifrequent alleles (thus high minor allele frequencies) are most informative for relatedness estimation [4]. The first SNP set (estimated minor allele frequencies > 0.35) aligns with prior and ongoing efforts to design other barcodes for malaria surveillance [17, 25].

Given evidence that a selective sweep on genes linked to artemisinin and piperaquine resistance (*kelch13* and *plasmepsin2–3*, respectively) substantially altered *P. falciparum* population structure across the GMS after 2012 [26–28], we included only sequences from 2009–2011 and excluded all sequences carrying the *kelch13* haplotype associated with this selective sweep (called KEL1 and identified using the haplotype scoring system in [28]). Lastly, we excluded two Pf3k study locations (Sisakhet and Ranong, Thailand) from the analysis that included < 10 *P. falciparum* individual sequences after removing post-2011 sequences and KEL1 mutants. Fig B in S1 Appendix provides information on the number of individual samples included from each location. To evaluate whether estimated gene flow decays with distance between locations, we obtained shortest road distances between Pf3k study locations from the Google Maps Distance Matrix API [29]. Between location distances ranged from 34 km (Bu Gia Map-Phouc Long) to 1783 km (Bu Gia Map-Bago Division).

## Mobility-informed coalescent simulation

We used a multi-population coalescent simulation to extend our analysis to a wider range of possible population genetic and epidemiological conditions. This approach also allows for unbiased sampling of the simulated populations, avoiding issues with sampling bias inherent to real sequence data collected in the field (as discussed in more detail below). We simulated sequence data for a metapopulation with five demes using the msprime implementation of Hudson's ms coalescent simulator [30, 31]. For consistency with the terms used to describe the Pf3k data, we hereafter refer to populations in the coalescent simulation as "locations". Here "individual" refers to either sequence data obtained from a single monoclonal *P. falciparum* infection in the Pf3k data or sequence data from a single individual in the simulations. Our analysis uses two coalescent modeling frameworks. Motivated by observations from existing studies [5, 32], and in our analysis of the Pf3k dataset, these models seek to approximate situations where migration patterns and effective population sizes vary over time, resulting in different population genetic signatures from ancestral versus more recent migration events. The first framework (Model A, Fig C in S1 Appendix) approximates an ancestral metapopulation with high mixing between locations, followed by a more recent period with lower migration rates with higher statistical variation. Specifically, Model A incorporates two different sets of migration rates between locations ($\rho_{i,j}$), an ancestral set in which migration is high and equal for all location pairs followed at time = $g$ generations in the past by a more recent set that is specified using aggregated, anonymized call detail records (CDRs) from mobile phone users in Thailand ($\rho_{i,j}^{CDR}$). We estimated migration rates (measured as the proportion of the population in $j$ that are migrants in $i$ per generation) between 930 districts in Thailand (Supplementary Methods in S1 Appendix) and sampled sets of districts over two spatial scales: "local" or nearby district sets separated by short geographic distances (approximately 100 km or less) and "subnational' district sets separated by larger geographic distances (Figs D, E, and F in S1 Appendix). We used sets of five districts each, sampling 10 local district sets and 10 subnational district sets, and used the resulting 5 × 5 migration rate matrices to parameterize the coalescent simulation. Effective population size is constant in Model A and denoted $N_f$.

The second framework (Model B, Fig C in S1 Appendix) seeks to approximate a metapopulation with recent-onset contraction in effective population sizes (for example, following successful implementation of malaria control interventions [32]). Model B uses the same mobility

data to specify a constant set of migration rates between locations, with an exponential decrease in effective population size from initial size ($N_i$) to current size ($N_f$) starting at time = $g$ generations in the past. The total number of migrants in each generation is a function of both migration rate and population size and contraction of effective population size reduces the total number of migrants starting after time = $g$.

For each simulation, we examine a range of parameters for $N_f$, $N_i$, $g$, and the recombination rate $\phi$. The mutation rate was set at 6.82E-9 mutations per site per generation for all simulations [19, 33]. We obtained a total of 1000 simulation replicates for each parameter set (100 simulation replicates for each of 10 sets of CDR-estimated migration rates). We calculated estimated allele frequencies for each site in the simulated data from a random sample of 80% of all individual sequences. Subsequent analysis was restricted to sites with estimated minor allele frequency > 0.35.

### Estimating gene flow between malaria parasite populations

Following [4], we define relatedness, $r$, between two individuals as the probability that, at any locus across the genome, the alleles for both individuals are IBD. In this study, we use the `fract_sites_IBD` output from *hmmIBD* [34] to estimate $r$ and let $\hat{r}$ denote this inter-individual relatedness estimate. To summarize numerous relatedness estimates between individuals from different locations, let $\hat{r}_\alpha$ denote the proportion of between-location estimates greater than some specified threshold, $\alpha$. We estimate inter-location differentiation using Hudson's estimator of $F_{ST}$ [35] and let $\hat{F}_{ST}$ denote this estimate of inter-location differentiation. We refer to $\hat{r}_\alpha$ and $\hat{F}_{ST}$ as estimates of gene flow since we expect gene flow between malaria parasite populations to correlate with both (positively with $\hat{r}_\alpha$ and negatively with $\hat{F}_{ST}$).

We calculate $\hat{r}_\alpha$ and $\hat{F}_{ST}$ using different numbers of individuals per location, $n$, and different numbers of SNPs, $p$. For the Pf3k data, we calculate $\hat{F}_{ST_{NP}}$ and $\hat{r}_{\alpha_{NP}}$ using all 437 SNPs with minor allele frequency estimates > 0.35 and, for each location pair, the maximum number of individuals available for each location. To distinguish IBD sharing due to ancestral versus more recent migration events, we also examine mean shared IBD sequence length for between-location sequence pairs and stratify this analysis over IBD segments of different lengths [36, 37]. Shorter IBD segment lengths, reflecting haplotypes that have been broken down by recombination over time, represent more distant ancestral events; longer IBD segments reflect sharing due more recent migration events.

### Ranking, classification, and isolation by distance for gene flow estimates

We calculate Kendall's $\tau$ for the correlation between $\hat{F}_{ST}$ or $\hat{r}_\alpha$ and geographic distance between locations, using distance between study cities for the Pf3k and distance between district centroids in Thailand for the simulated data (which uses administrative districts as the unit of aggregation for mobile phone-associated movement data). A Mantel test with 1000 permutations was used to evaluate the statistical significance of each $\tau$ value.

For the simulated data, where the migration rate between locations is known *a priori*, we compared differences in estimated gene flow to differences in pre-specified migration rates to determine whether $\hat{F}_{ST}$ or $\hat{r}_\alpha$ values correctly rank pairs of location pairs. For example, if a location pair with a high migration rate has a higher estimated value for $\hat{r}_\alpha$ than a location pair with a lower pairwise migration rate, we consider these two location pairs as ranked correctly by their respective $\hat{r}_\alpha$ values. We calculate the proportion of all 45 pairs of location pairs (five choose two choose two) that are ranked correctly by their $\hat{F}_{ST}$ or $\hat{r}_\alpha$ values (proportion ranked correctly, "PRC"). We examine PRC as a practical measure for contextualizing observed values

of $\hat{F}_{ST}$ or $\hat{r}_{\alpha}$ (i.e. if $n$ total individuals are sampled and genotyped across $p$ SNPs, what is the probability that the observed values for $\hat{F}_{ST}$ or $\hat{r}_{\alpha}$ correctly rank location pairs by their shared gene flow?).

Motivated by expected practical use cases in malaria genomic surveillance, we also consider how well $\hat{F}_{ST}$ or $\hat{r}_{\alpha}$ identify or classify highly connected location pairs. Specifically, for each individual simulation, we evaluate whether (1) the single location pair with the highest migration rate also has the highest estimated pairwise gene flow and (2) whether the location pairs with the five highest migration rates also have the five highest gene flow estimates. For each parameter set, we report the proportion of all simulations in which these location pairs are classified correctly.

## Results

### Relatedness and differentiation between *P. falciparum* populations

We first estimated connectivity between *P. falciparum* sequences collected in multiple study sites across the Greater Mekong Subregion using both differentiation-based and relatedness-based estimates of gene flow ([Fig 1]). We observed distinct signatures of isolation by distance with both $\hat{F}_{ST_{NP}}$ (increasing differentiation with distance) and $\hat{r}_{0.1_{NP}}$ and $\hat{r}_{0.5_{NP}}$ (decreasing relatedness with distance). The strength of these correlations decreases with smaller data sizes, most markedly with $p$ (the number of SNPs used) $< 100$, and by SNP ascertainment scheme ([Fig 2]). Using SNPs with higher estimated minor allele frequencies ($> 0.35$) yielded stronger correlations with distance for both $\hat{F}_{ST_{NP}}$ and $\hat{r}_{0.5_{NP}}$; using SNPs from a wider range of estimated minor allele frequencies ($> 0.05$), and thus including more rare alleles, yielded less consistent correlations with distance.

Examining location pairs with the highest estimated shared gene flow, we found largely similar geographic patterns for $\hat{F}_{ST_{NP}}$ and $\hat{r}_{0.1_{NP}}$ ([Fig 1A and 1B]), with the strongest estimated gene flow observed between geographically-proximate locations in southern Laos, eastern Cambodia, and Vietnam. However, $\hat{r}_{0.5_{NP}}$ values were highest (consistent with higher gene flow) for location pairs within southern Cambodia and Vietnam. We found an identical geographic pattern when examining mean pairwise IBD sequence length for long IBD tracts ($> 95^{th}$ percentile, equal to tract length $\geq 285.64$ kb), indicating that these patterns reflect more recent migration events (Fig G in [S1 Appendix]) [36, 37]. Location pairs with the highest $\hat{r}_{0.5_{NP}}$ values also have relatively higher numbers of nearly clonal individual-individual pairs (Fig H in [S1 Appendix]), which result from recent migration events (where the time since migration is short enough that there is limited or no out-crossing between imported individuals and the receiving population). Mean IBD segment length for between-location individual-individual pairs is also longer for location pairs with the highest $\hat{r}_{0.5_{NP}}$ (Fig H in [S1 Appendix]). These findings indicate that population-level summaries of between-location relatedness, if used with appropriate thresholds that can identify highly-related individual-individual pairs, can provide useful estimates of recent migration between *P. falciparum* populations. $\hat{F}_{ST}$ reflects gene flow due to both recent and more historically distant migration events and in some contexts may be strongly influenced by more ancestral population structure.

Sequence data in the Pf3k database was obtained during multiple clinical and cross-sectional studies [38], each of which may be subject to different forms of sampling bias. We observed marked heterogeneity across study sites in the within-location distributions for $\hat{r}$ (Fig I in [S1 Appendix]). The proportion of clonal or nearly-clonal individual-individual pairs (where $\hat{r} \approx 1$) differs widely across study locations, likely reflecting both true differences in

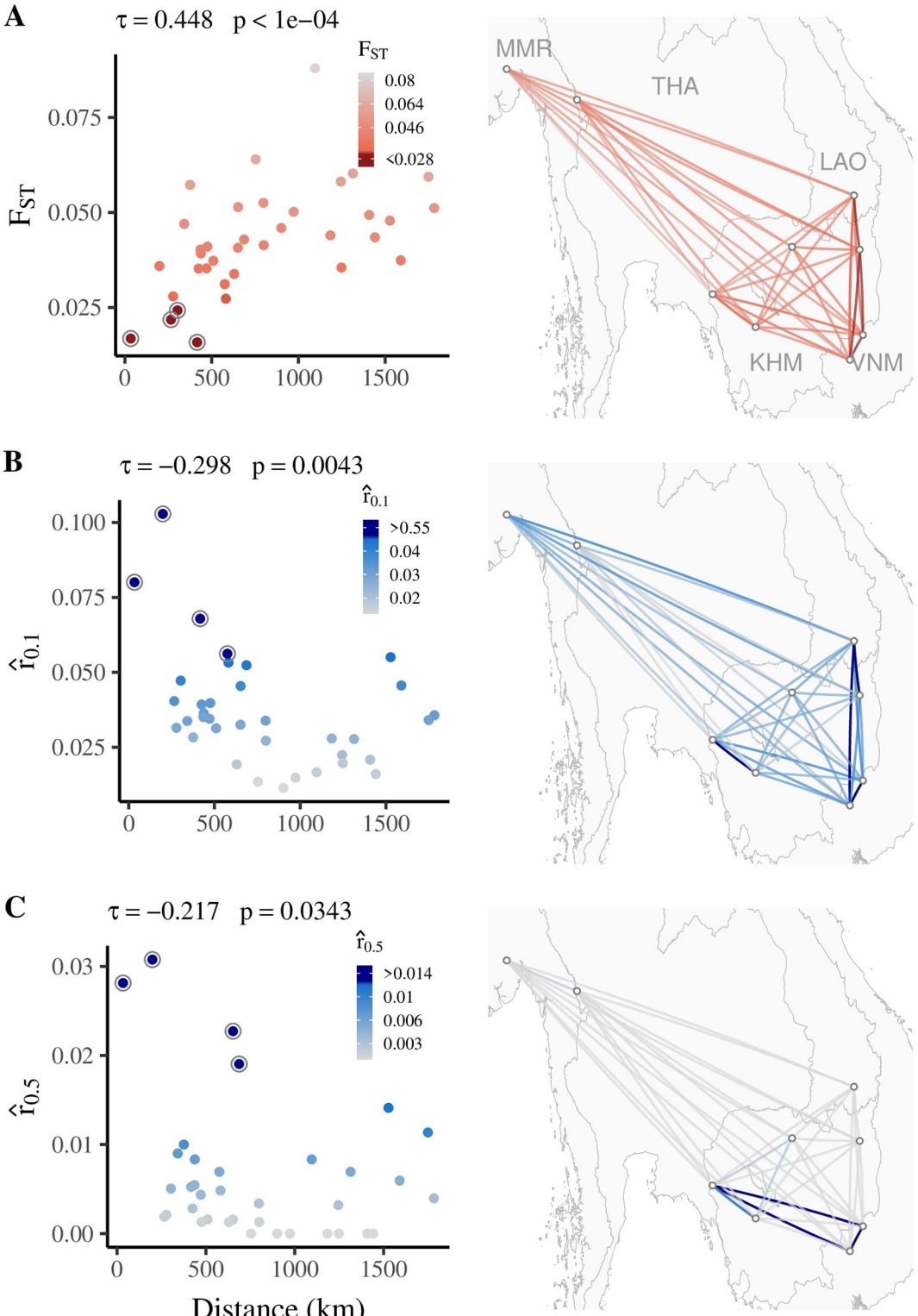

**Fig 1. Gene flow versus distance for *P. falciparum* isolates.** (A) $F_{ST}$ (Hudson's estimator), (B) $\hat{r}_{0.1}$, and (C) $\hat{r}_{0.5}$. Kendall's $\tau$ for divergence or relatedness versus distance and the corresponding p-value obtained via Mantel testing are listed for each metric. The four location pairs with highest estimated gene flow (as measured by $F_{ST}, \hat{r}_{0.1}$, or $\hat{r}_{0.5}$) are circled in the left panels. Contains information from OpenStreetMap and OpenStreetMap Foundation, which is made available under the Open Database License.

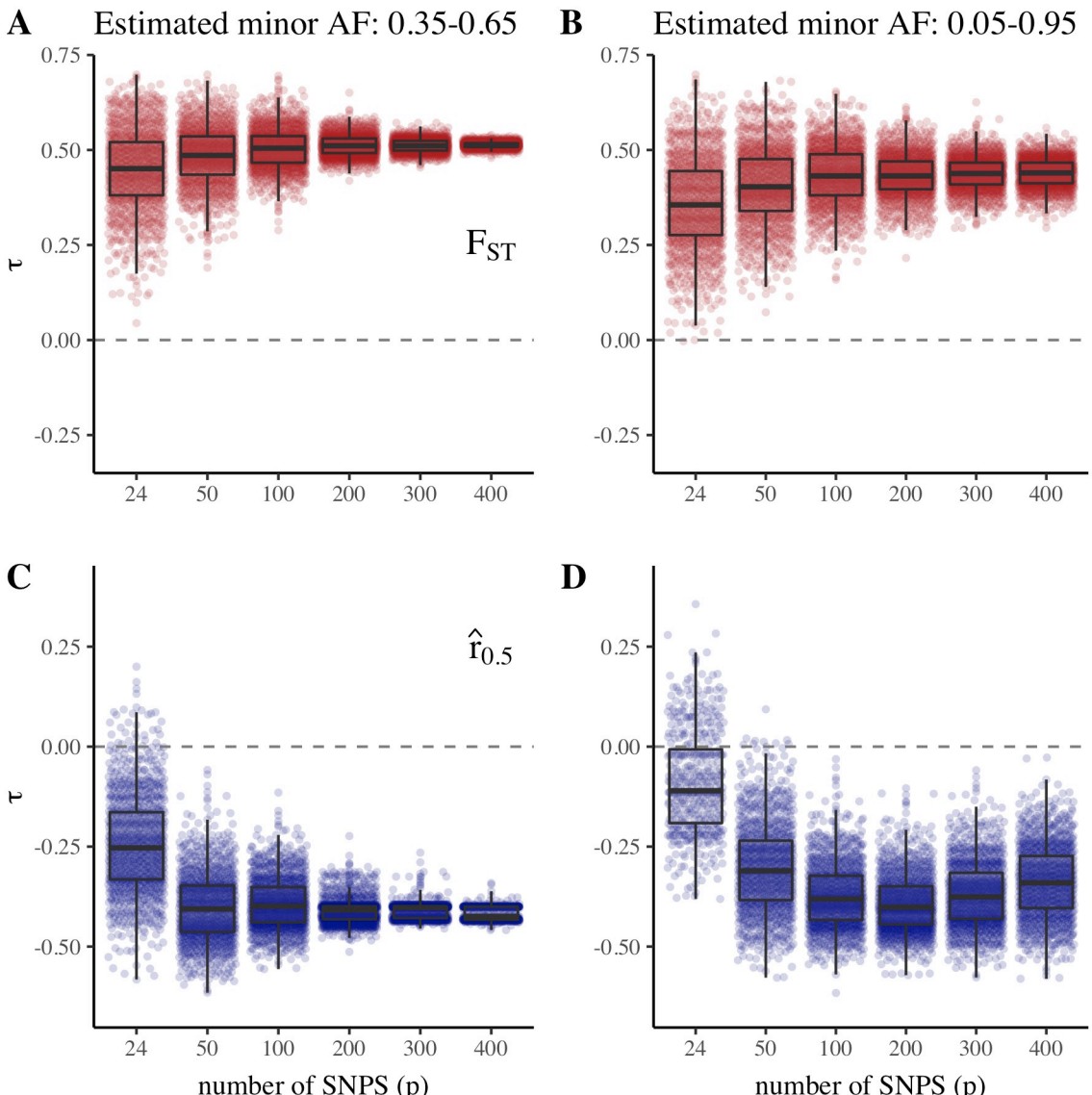

**Fig 2. Gene flow versus distance for *P. falciparum* isolates by SNP number and estimated minor allele frequency (AF).** Kendall's $\tau$ for distance versus $F_{ST}$ (A & B) or $\hat{r}_{0.5}$ (C & D) for SNPs with estimated minor allele frequency of $> 0.35$ (A & C) or $> 0.05$ (B & D). Points show $\tau$ values for 1000 SNP sets of size $p$ randomly sampled from the subsets of SNPs with estimated minor allele frequency of either $> 0.35$ or $> 0.05$.

levels of within-population diversity and/or sampling bias (for example, if samples were collected from individuals infected with the same clone during an outbreak or other shared transmission event).

## Data size and isolation by distance over local and subnational scales

To extend this analysis, we used a mobility-informed coalescent simulation to numerically evaluate $\hat{F}_{ST}$ and $\hat{r}_{\alpha}$ under a wider range of possible epidemiological and evolutionary conditions. This approach also allows for unbiased sampling of individuals in each location,

obviating the aforementioned issues with sampling bias in real *P. falciparum* sequence data sets. The two models used for this purpose (Fig C in S1 Appendix, Models A and B) yield $\hat{F}_{ST}$ and $\hat{r}_{0.5}$ values that closely approximate those observed for *P. falciparum* populations over both subnational and local geographic scales (Figs J, K, and L in S1 Appendix) [5, 38]. Simulations with higher recombination rates better approximated observed $\hat{F}_{ST}$ and $\hat{r}_{0.5}$ vales in the Pf3k dataset (Figs M and N in S1 Appendix). The CDR-derived mobility data used to parameterize these models indicates that between-district migration in Thailand is less strongly correlated with distance over local geographic areas (i.e. between nearby districts) compared to subnational movement over a wider range of distances (Fig D in S1 Appendix). This suggests that migration may be less spatially coherent over short distances, such that the presence or absence of isolation by distance may be difficult to interpret as a signature of population structure under certain conditions.

Consistent with this finding, we observe weaker correlations between geographic distance and both $\hat{F}_{ST}$ and $\hat{r}_{0.5}$ for migration over local scales (Fig 3) when compared to migration over larger subnational scales (Fig 4). Similarly, $\hat{F}_{ST}$ and $\hat{r}_{0.5}$ are more closely correlated with migration rates (as specified in the coalescent model) than distance (Figs O and P in S1 Appendix). These findings are in part attributable to the wider variation of distances, and larger differences in migration rates, for the districts sampled to represent subnational migration, which include districts separated by both large and small distances (Figs D and F in S1 Appendix). Migration rates between nearby or local districts are higher overall and restricted to a narrower range of values with lower dispersion, resulting in $\hat{F}_{ST}$ and $\hat{r}_{0.5}$ values that are less strongly correlated with distance (Figs J and K in S1 Appendix).

We next examined how the ability to detect isolation by distance is influenced by data size. To do this, we evaluated the proportion of all simulation replicates, of data size of $p$ SNPs and $n$ individuals, for which the correlation between distance and $\hat{F}_{ST}$ or $\hat{r}_{0.5}$ is significant by Mantel testing. These proportions (proxy measures for the power to detect isolation by distance) are markedly decreased for data sizes with $< 100$ SNPs or $< 50$ individuals, across both models A and B and for both local and subnational migration rates (Figs 3 and 4 and Fig Q in S1 Appendix).

In both models A and B, we found $\hat{r}_{0.5}$ identified isolation by distance more reliably than $\hat{F}_{ST}$ over almost all data sizes (Figs 3 and 4 and Fig Q in S1 Appendix). This likely reflects the relatively short number of post-ancestral generations used in our simulations, in which the change from "ancestral" to "recent" migration rates (Model A) or the onset of population size contraction (Model B) occurs at 10–50 generations before present. These findings are consistent with recent studies on the Thai-Myanmar border, where contraction of *P. falciparum* effective population size began approximately 20 years ago [32] (80–120 generations, assuming 4–6 generations per year) and where IBD-based methods for estimating gene flow delineated hyperlocal spatial population structure that could not be resolved by $F_{ST}$ [5]. Supporting this observation, sensitivity analysis examining different values for $g$ (in Model A, the number of generations before present when ancestral migration rates are supplanted by recent migration rates) indicates that $\hat{r}_{0.5}$ values are strongly influenced by $g$, whereas $F_{ST}$ values are largely similar for both large and small values of $g$ (Fig R in S1 Appendix).

## Data size and ranking gene flow estimates

Motivated by practical questions in malaria surveillance, and acknowledging the potential limitations of isolation by distance in contexts where migration is poorly correlated with distance, we sought to evaluate data size requirements for ranking and classification of gene flow

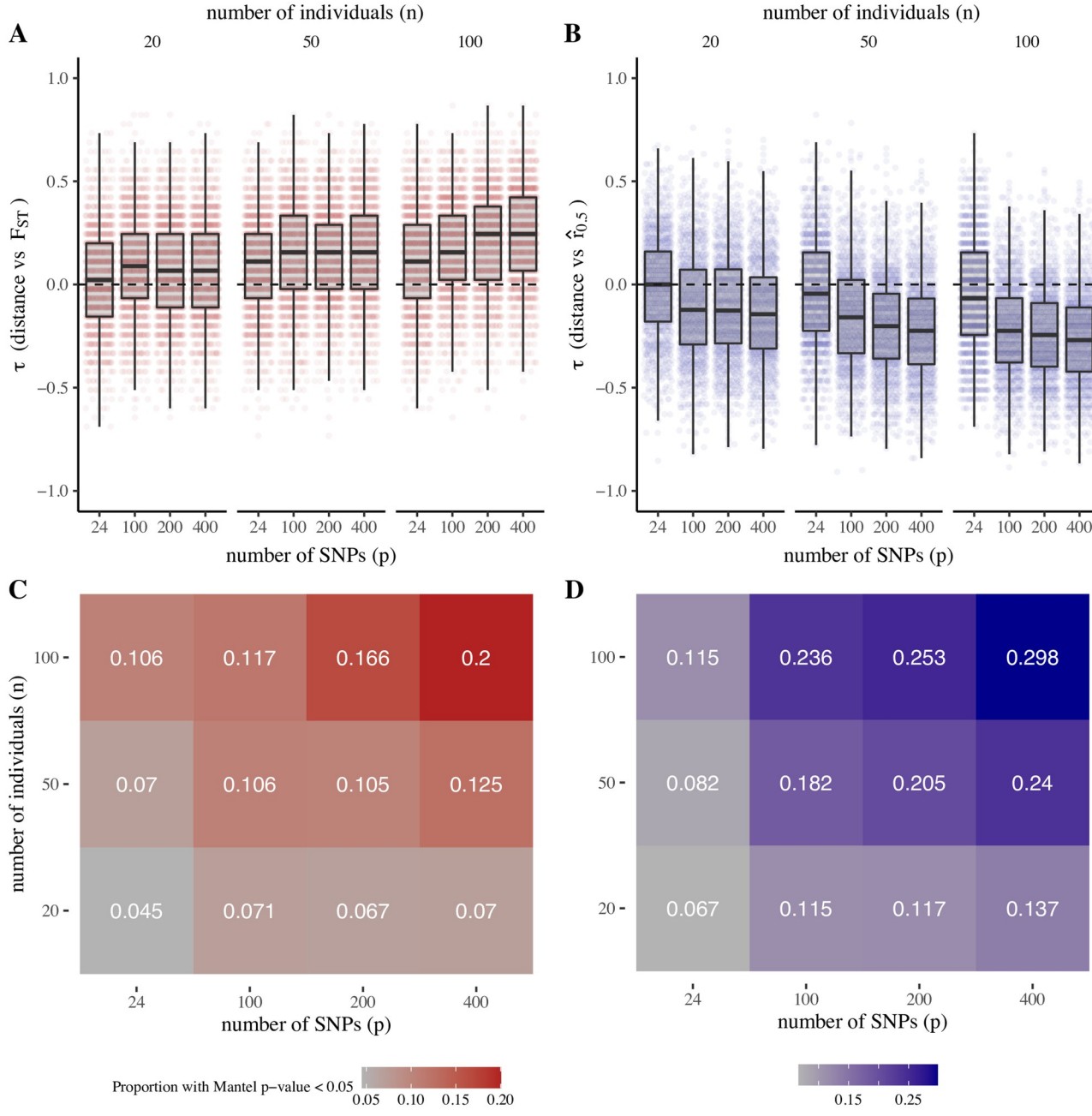

**Fig 3. Gene flow versus distance for simulated sequence data (local migration, model A).** Migration rates are estimated using CDR data for neighboring districts in Thailand (separated by $\geq$ 100 km). (A) and (B): correlation between distance and observed values of $F_{ST}$ and $\hat{r}_{0.5}$, respectively. Points show $\tau$ values for 1000 independent simulation replicates. (C) and (D): proportion of simulation replicates where Mantel-estimated p-values for observed $\tau$ values are $\geq$ 0.05, by number of SNPs ($p$) and number of individuals ($n$). Model parameters: recombination rate, $\phi = 0.7$, multiplier for recent migration rates, $m = 15$; multiplier for ancestral migration rates, $M = 5$; population size, $N_f = 500$; time since ancestral migration rates, $g = 10$ generations.

estimates. To examine ranking, we calculated for each simulation replicate the proportion of all pairs of locations where ranking by $\hat{F}_{ST}$ or $\hat{r}_{0.5}$ matches ranking according to the migration rates specified in the coalescent simulation ("proportion ranked correctly"). This proportion approximates the probability, for a given simulation replicate with data size $p$ SNPs and $n$

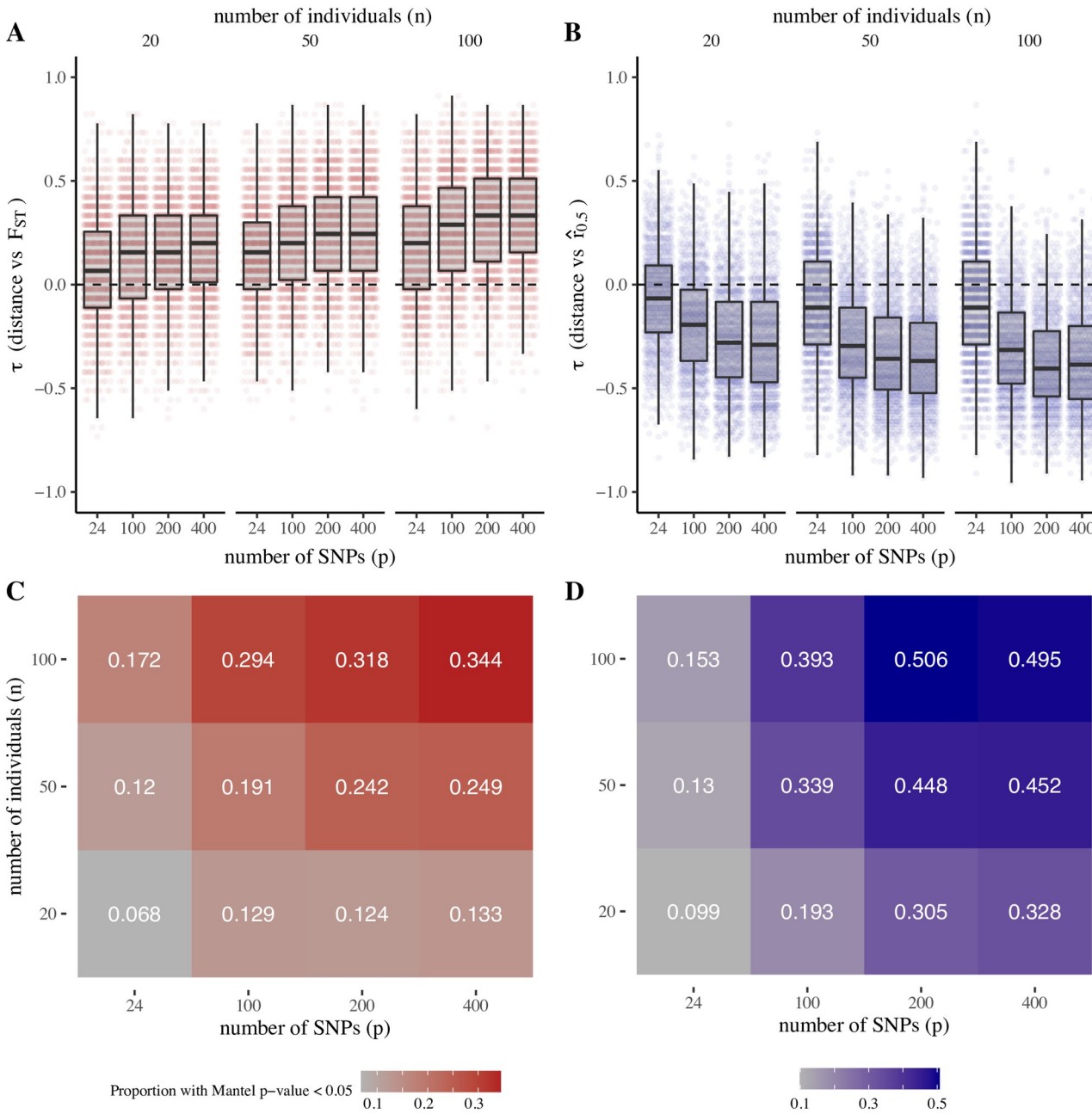

**Fig 4. Gene flow versus distance for simulated sequence data (subnational migration, model A).** Migration rates are estimated using CDR data for neighboring districts in Thailand. Distances are (A) and (B): correlation between distance and observed values of $F_{ST}$ and $\hat{r}_{0.5}$, respectively. Points show $\tau$ values for 1000 independent simulation replicates. (C) and (D): proportion of simulation replicates where Mantel-estimated p-values for observed $\tau$ values are $\geq 0.05$, by number of SNPs ($p$) and number of individuals ($n$). Model parameters: recombination rate, $\phi = 0.7$, multiplier for recent migration rates, $m = 15$; multiplier for ancestral migration rates, $M = 5$; population size, $N_f = 500$; time since ancestral migration rates, $g = 10$ generations.

individuals, that the $\hat{F}_{ST}$ or $\hat{r}_{0.5}$ values for two location pairs will correspond to the true ordering of their respective migration rates. Overall, a lower proportion of location pairs were ranked correctly when migration is specified using local (Fig S in S1 Appendix) rather than subnational migration matrices (Fig T in S1 Appendix). Notably, when migration is specified

using subnational district sets, the proportion of location pairs ranked correctly by $\hat{F}_{ST}$ is greater than the proportion ranked correctly by $\hat{r}_{0.5}$ over almost all data sizes (Fig T in S1 Appendix).

Lastly, we examined, for a given data size, the ability to correctly distinguish highly connected location pairs from less highly connected pairs using $\hat{F}_{ST}$ or $\hat{r}_{0.5}$. We found that misclassification of highly connected location pairs was more frequent when local rather than subnational district sets were used to specify migration (Figs S and T in S1 Appendix). The five location pairs with highest connectivity were correctly classified in only a small proportion of simulation replicates (Figs S and T in S1 Appendix), indicating that distinguishing highly connected groups of location pairs may be difficult in many epidemiological contexts.

## Discussion

Genomic and genotype-based methods have become increasingly important tools for malaria surveillance and control, providing valuable information on the impact of disease control interventions [20], the geographic dispersal of antimalarial drug resistance [36], and connectivity between regional and local parasite populations [5], among other critical insights. There is an ongoing effort to operationalize these tools for use in routine disease surveillance activities, and developing the evidence base around their optimal use is a central scientific and public health priority. In this study, we used genome sequence data from naturally-occurring *P. falciparum* infections and sequence data simulated using parameters associated with different epidemiological conditions to examine a specific but potentially important use cases in malaria genomic surveillance, i.e. testing for isolation by distance and ranking gene flow between different *P. falciparum* population pairs.

We highlight the following observations:

1. Recent and ancestral population dynamics, together with the magnitude and dispersion of migration rates, all influence the ability to distinguish levels of gene flow between malaria populations.

2. Identifying highly related or nearly clonal between-location sequence pairs provides an important measure for recent parasite migration events. IBD-based measures that capture these recent migration events (including $\hat{r}_{\alpha}$) are less influenced by ancestral migration than $F_{ST}$.

3. The ability to identify isolation by distance and rank gene flow is lower when 24-SNP datasets are used or when fewer than 50 individuals are sampled per location. Larger data sizes (100 or more SNPs and $> 50$ individuals per location) are recommended.

4. SNPs with higher estimated minor allele frequencies more reliably identify isolation by distance. For $\hat{r}_{\alpha}$, these results are consistent with prior work [4]. $F_{ST}$ estimates can be biased by the allele frequencies of the marker set under analysis, with the inclusion of more rare polymorphisms leading to under-estimation of $F_{ST}$ and enrichment for more equifrequent polymorphisms resulting in over-estimation [39]. Bias toward lower $F_{ST}$ estimates when using the SNP set with more rare alleles (estimated minor allele frequency $> 0.05$) may explain why this ascertainment scheme yielded weaker correlations between $F_{ST}$ and distance (when compared with the set of more equifrequent SNPs). Identifying optimal SNP ascertainment strategies for malaria genomic surveillance studies is an area of active research [16].

5. Detecting or failing to detect isolation by distance may be less informative if parasite migration rates are not consistently correlated with distance [40], as observed in our analysis of

mobile phone-associated movement data (specifically, for migration between nearby or local districts in Thailand) and in other studies [8–10]. In these contexts, the ability to reliably rank gene flow estimates or distinguish highly connected versus less highly connected location pairs may be more useful from a programmatic standpoint.

6. Reliable ranking of gene flow between pairs of populations may be difficult in situations where migration rates are high or highly uniform across location pairs (for example, between local or nearby districts), even when using larger numbers of highly informative SNPs and sampling large numbers of individuals.

There are multiple limitations that are important for understanding these observations in context. The *P. falciparum* sequence data we obtained from the Pf3k database includes only a small number of individuals from certain locations (for example, Pailin, Cambodia and Bago Division, Myanmar). This analysis is limited to samples collected over a two-year period (2009–2011) in a relatively small number of locations in Southeast Asia, and observations from this dataset are not generalizable beyond the epidemiological and ecological conditions specific to this context. In addition, our analysis is likely influenced by biases inherent to sample collection in each Pf3k study location, making it difficult to disambiguate whether observed patterns in relatedness and differentiation are due to underlying *P. falciparum* population structure or sampling biases. Potential sampling biases in field studies include over-sampling of clinical or outbreak related infections, with resultant undersampling of subclinical infections, and laboratory-based sources of bias related to difficulties genotyping infections with low parasite density. Our analysis also excluded polyclonal infections and individuals carrying resistance-associated KEL1 haplotypes. Removal of polyclonal infections may result in substantial levels of missingness, particularly in highly endemic contexts where polyclonal infections are more common. Moreover, the distribution of polyclonal infections across locations may provide important information about gene flow between malaria populations. Likewise, exclusion of individuals carrying known resistance-associated mutations may result in substantial loss of potentially informative data and focusing on these isolates may provide insights into patterns of connectivity underlying dispersal of drug resistance-associated genetic polymorphisms. Lastly, it is not clear that geographic distance between Pf3k study locations should correlate with parasite migration rates and signatures consistent with isolation by distance should be interpreted with caution here.

We sought to address some of these limitations via coalescent simulation, allowing for numerical evaluation of relatedness- and differentiation-based approaches over a wider range of possible epidemiological conditions. This approach avoids the potential biases inherent to sampling from naturally-occurring *P. falciparum* populations in the field and allows for direct comparison between known migration rates between locations and their corresponding $F_{ST}$ and $\hat{r}_\alpha$ values. Parameters used for coalescent simulation are subject to potential mis-specification, particularly those parameters where there is uncertainty about their true values in naturally-occurring *P. falciparum* populations (for example, recombination rate). Although we do not statistically infer model parameters, the parameters used in our simulations yield $F_{ST}$ and $\hat{r}_\alpha$ values that are consistent with the range and distribution of values observed in Pf3k data. Importantly, we specified equal population sizes across the five locations in the coalescent simulation, despite the fact that differential transmission between locations (resulting in larger and smaller effective population sizes) could have important impacts on gene flow estimates.

There are other aspects of our approach that may limit generalizability to real *P. falciparum* populations in the field. For example, we have assumed the ability to sample individuals evenly across populations, and that sampling by default captures every population in the metapopulation; in real-life genomic surveillance applications, sampling is likely to be unbalanced and

imperfect, such that individuals from certain populations may be undersampled or not captured at all. In addition, our analysis only considers direct connectivity between locations and, in certain contexts, indirect migration between locations may be an important contributor to gene flow. Lastly, our analysis assumes perfect overlap of survey catchment area and population, whereas in real-life surveillance applications, it is more difficult to ascribe a known geolocation to each individual infection (given that individuals may move location between infection and medical diagnosis or survey participation) and thus the task of assigning individual infectious to specific locations involves much more uncertainty.

## Conclusion

In conclusion, we have examined numerically the properties of $\hat{F}_{ST}$ and $\hat{r}_\alpha$ as estimates of population-level gene flow between *P. falciparum* populations, and outlined, insofar as our simulations allow, basic data requirements (number of individuals and number of SNPs) for their use. Although this study is focused on a limited set of genomic surveillance applications (identifying isolation by distance and correctly ranking location pairs versus less highly connected ones), our work underscores an important limitation that applies across a wider range of surveillance applications: under certain epidemiological conditions (high migration between large populations) it likely becomes infeasible to reliably rank gene flow between *P. falciparum* populations, even with dense sampling of individuals and sequence data for large numbers of highly informative genetic markers. Additional scientific work is needed to understand how unbalanced sampling of individuals, incomplete sampling of metapopulations (for example, non-sampling of entire demes), and other exigencies of real-world genomic surveillance impact the ability to rank different levels of gene flow between *P. falciparum* populations in the field.

## Supporting information

**S1 Appendix. Supplementary methods and figures. Fig A. Positions for filtered SNPs from the Pf3k dataset**. (A) SNPs with estimated minor allele frequency > 0.35. (B) SNPs with estimated minor allele frequency > 0.05. **Fig B. Number of monoclonal *P. falciparum* sequences from the Pf3k database included in this study by location**. Data includes only those sequences collected between 2009 and 2011 and excludes KEL1 mutants (as described in Methods). Contains information from OpenStreetMap and OpenStreetMap Foundation, which is made available under the Open Database License. **Fig C. Coalescent models used to generate simulated sequence data**. (A): Constant population size with change from ancestral migration rates to current migration rates at time = $g$ generations in the past. The ancestral migration rate is equal for all location-location pairs and specified as $\rho_{ij} = M \times \max(\rho^{CDR})$, where $M$ is a multiplier. Recent migration rates are specified using CDR-estimated mobility data ($\rho^{CDR}$, as described in the Methods and Supplementary Methods) and a multiplier value $m$. (B): Constant migration rates with exponential decrease in population size from an initial size ($N_i$) to current size ($N_f$) starting at time = $g$ generations in the past. (C): Constant population size and constant migration rates. **Fig D. CDR-estimated migration rates**. (A) Relative migration rates ($\rho_{CDR}/\max(\rho_{CDR})$) between 930 districts in Thailand with adequate CDR data. (B) 10 randomly sampled sets of n = 5 neighboring districts with distances between district centroids of approximately 100 km or less. (C) 10 randomly sampled sets of n = 5 distantly-separated districts, selected to include districts > 600 km from randomly chosen a center district. Sampling procedures for the district sets in (B) and (C) and their use for specifying migration rates in the coalescent simulation are described in the Methods and Supplementary Methods. **Fig E. Example sets of nearby districts**. Figure shows four of ten total randomly

selected district sets used to specify migration rates in the coalescent model (corresponding to Panel B in Fig D). Contains information from OpenStreetMap and OpenStreetMap Foundation, which is made available under the Open Database License. **Fig F. Example sets of distant districts**. Panels show four of ten total randomly selected district sets used to specify migration rates in the coalescent model (corresponding to Panel C in Fig D). Contains information from OpenStreetMap and OpenStreetMap Foundation, which is made available under the Open Database License. **Fig G. Distance versus alternative population-level estimates of relatedness**. (A) Distance versus mean pairwise IBD sequence length when only the longest IBD tract lengths are considered ($> 95^{th}$ percentile, equal to tract length $\geq$ 285.64 kb). (B) Distance versus $\hat{r}_{0.8}$, the proportion between-location individual-individual pairs with $r > 0.8$. Contains information from OpenStreetMap and OpenStreetMap Foundation, which is made available under the Open Database License. **Fig H. Between-location relatedness for *P. falciparum* individual-individual pairs**. (A) Distribution of $r$ values (proportion of SNPs that are IBD between pairs) for between-location individual-individual pairs across 36 location-location pairs in the Great Mekong Subregion. (B) Distribution of mean pairwise IBD segment length (the mean length of shared IBD tracts for individual-individual pairs) for the same location-location pairs. Symbols show location-location pairs with the highest estimated gene flow per $F_{ST}$ (red circles), $\hat{r}_{0.1}$ (blue circles), $\hat{r}_{0.5}$ (open blue squares), and mean pairwise IBD sequence length (as described in Fig G, filled blue squares). **Fig I. Pairwise relatedness for intra-location individual-individual pairs**. Plots show distribution of intra-location $r$ values for 9 locations (proportion of SNPs that are IBD between within-location pairs of sequences) included from the Pf3k dataset. **Fig J. Observed values for $F_{ST}$ and $\hat{r}_{0.5}$ coalescent-simulated sequence data (local migration, model A)**. Data was simulated using constant population size and instantaneous change from ancestral migration rates to recent migrations at *time* = *g* generations in the past (as described in Panel A of Fig C). Model parameters: recombination rate, $\phi$ = 0.7, multiplier for recent migration rates, $m$ = 15; multiplier for ancestral migration rates, $M$ = 5; population size, $N_f$ = 500; time since ancestral migration rates, $g$ = 10 generations. (A) and (B): Observed $F_{ST}$ values compared to specified migration rates and distance between districts, respectively. (C) and (D): Observed $\hat{r}_{0.5}$ values compared to specified migration rates and distance between districts, respectively. Filled circles show values for 1000 independent simulations using 10 randomly sampled district sets specifying CDR-estimated migration rates (as shown in Fig D). Open circles show the mean $F_{ST}$ or $\hat{r}_{0.5}$ values for each migration rate or distance. Rug plot on y-axis shows estimated $F_{ST}$ or $\hat{r}_{0.5}$ values obtained from the Pf3k *P. falciparum* data (for comparison with the model-derived values). **Fig K. Observed values for $F_{ST}$ and $\hat{r}_{0.5}$ coalescent-simulated sequence data (subnational migration, model A)**. Data was simulated using constant population size and instantaneous change from ancestral migration rates to recent migrations at *time* = *g* generations in the past (as described in Panel A of Fig C). Model parameters: recombination rate, $\phi$ = 0.7, multiplier for recent migration rates, $m$ = 15; multiplier for ancestral migration rates, $M$ = 5; population size, $N_f$ = 500; time since ancestral migration rates, $g$ = 10 generations. (A) and (B): Observed $F_{ST}$ values compared to specified migration rates and distance between districts, respectively. (C) and (D): Observed $\hat{r}_{0.5}$ values compared to specified migration rates and distance between districts, respectively. Filled circles show values for 1000 independent simulations using 10 randomly sampled district sets specifying CDR-estimated migration rates (as shown in Fig D). Open circles show the mean $F_{ST}$ or $\hat{r}_{0.5}$ values for each migration rate or distance. Rug plot on y-axis shows estimated $F_{ST}$ or $\hat{r}_{0.5}$ values obtained from the Pf3k *P. falciparum* data (for comparison with the model-derived values). **Fig L. Observed values for $F_{ST}$ and $\hat{r}_{0.5}$ coalescent-simulated sequence data (local migration, model B)**. Data was simulated using a coalescent model with a constant set of

migration rates and exponential decrease from an ancestral population size ($N_i$) to current populations size ($N_f$) starting *time = g* generations in the past (as described in Panel B of Fig C). Migration rates are specified using CDR-estimated mobility between nearby districts in Thailand separated by approximately 100 km or less ("local" migration). Model parameters: recombination rate, $\phi = 0.7$, multiplier for migration rates, $m = 15$; final population size, $N_f = 100$; ancestral population size, $N_i = 1000$; time since onset of exponential decrease in population size, $g = 50$ generations. (A) and (B): Observed $F_{ST}$ values compared to specified migration rates and distance between districts, respectively. (C) and (D): Observed $\hat{r}_{0.5}$ values compared to specified migration rates and distance between districts, respectively. Filled circles show values for 1000 independent simulations using 10 randomly sampled district sets specifying CDR-estimated migration rates (as shown in Fig D). Open circles show the mean $F_{ST}$ or $\hat{r}_{0.5}$ values for each migration rate or distance. Rug plot on y-axis shows estimated $F_{ST}$ or $\hat{r}_{0.5}$ values obtained from the Pf3k *P. falciparum* data (for comparison with the model-derived values). **Fig M. Sensitivity analysis for recombination rate $\phi$ used in coalescent simulations (local migration, model A)**. Model parameters: recombination rate, $\phi = 0.1$ or $0.7$, multiplier for recent migration rates $m = 15$, multiplier for ancestral migration rates $M = 5$; final population size, $N_f = 500$; time since ancestral migration rates, $g = 10$ generations. (A) and (B): Observed $F_{ST}$ values compared to specified migration rates between districts for $\phi = 0.1$ and $0.7$, respectively. (C) and (D): Observed $\hat{r}_{0.5}$ values compared to specified migration rates. Filled circles show values for 1000 independent simulations using 10 randomly sampled district sets specifying CDR-estimated migration rates. Open circles show the mean $F_{ST}$ or $\hat{r}_{0.5}$ values for each migration rate or distance. Rug plot on y-axis shows estimated $F_{ST}$ or $\hat{r}_{0.5}$ values obtained from the Pf3k *P. falciparum* data (for comparison with the model-derived values). **Fig N. Sensitivity analysis for recombination rate $\phi$ used in coalescent simulations (local migration, model B)** Model parameters: recombination rate, $\phi = 0.1$ or $0.7$, multiplier for migration rates $m = 15$; final population size, $N_f = 100$; ancestral population size, $N_i = 1000$; time since onset of exponential decrease in population size, $g = 50$ generations. (A) and (B): Observed $F_{ST}$ values compared to specified migration rates between districts for $\phi = 0.1$ and $0.7$, respectively. (C) and (D): Observed $\hat{r}_{0.5}$ values compared to specified migration rates. Filled circles show values for 1000 independent simulations using 10 randomly sampled district sets specifying CDR-estimated migration rates. Open circles show the mean $F_{ST}$ or $\hat{r}_{0.5}$ values for each migration rate or distance. Rug plot on y-axis shows estimated $F_{ST}$ or $\hat{r}_{0.5}$ values obtained from the Pf3k *P. falciparum* data (for comparison with the model-derived values). **Fig O. Gene flow versus migration rate for coalescent-simulated sequence data (local migration, model A)**. Migration rates are estimated using CDR data for neighboring districts in Thailand (separated by $\leq$ 100 km). (A) and (B) correlation between migration rate (as specified in the coalescent model) and observed values of $F_{ST}$ and $\hat{r}_{0.5}$, respectively. Points show $\tau$ values for 1000 independent simulation replicates. (C) and (D): proportion of simulation replicates where Mantel-estimated p-values for observed $\tau$ values are $\geq 0.05$, by number of SNPs ($p$) and number of individuals ($n$). **Fig P. Gene flow versus migration rate for coalescent-simulated sequence data (subnational migration, model A)**. Migration rates are estimated using CDR data for distant districts in Thailand. (A) and (B) correlation between migration rate (as specified in the coalescent model) and observed values of $F_{ST}$ and $\hat{r}_{0.5}$, respectively. Points show $\tau$ values for 1000 independent simulation replicates. (C) and (D): proportion of simulation replicates where Mantel-estimated p-values for observed $\tau$ values are $\geq 0.05$, by number of SNPs ($p$) and number of individuals ($n$). **Fig Q. Gene flow versus distance for simulated sequence data (local migration, model B)** Data was simulated using a coalescent model with a constant set of migration rates and exponential decrease from an ancestral population size ($N_i$) to current

populations size ($N_f$) starting *time* = *g* generations in the past (as described in Panel B of Fig C). Migration rates are estimated using CDR data for neighboring districts in Thailand (separated by $\geq$ 100 km). (A) and (B): correlation between distance and observed values of $F_{ST}$ and $\hat{r}_{0.5}$, respectively. Points show $\tau$ values for 1000 independent simulation replicates. (C) and (D): proportion of simulation replicates where Mantel-estimated p-values for observed $\tau$ values are $\geq$ 0.05, by number of SNPs (*p*) and number of individuals (*n*). **Fig R. Time since ancestral migration events versus $\hat{F}_{ST}$ or $\hat{r}_{0.5}$ (Model A)**. Top panels: Each point compares the $\hat{F}_{ST}$ value for the same location pair in the simulation using *g* = 10 (x-axis) versus *g* = 500 (y-axis). Bottom panels: Model parameters: *M*, the multiplier for ancestral migration rates (before *g* generations in the past), is equal to either 5 (left panels) or 50 (right panels). *m*, the multiplier for recent migration rates, is equal to 15 and the recombination rate, $\phi$, is 0.7 for all simulations shown. Points are colored by the number of migrants per generation in the "recent" migration matrix (from 0 to *g* generations in the past). The dashed line shows where x-axis and y-axis values are equal. **Fig S. Ranking and classification metrics for coalescent-simulated sequence data (local migration, model A)** (A) and (B): Proportion of all location-location pairs that are ranked correctly by either $F_{ST}$ or $\hat{r}_{0.5}$, respectively, when compared to the migration rate specified in the coalescent simulation. Points show these values for 1000 independent simulation replicates over different numbers of individuals (*n*) and SNPs (*p*) used to calculate $F_{ST}$ or $\hat{r}_{0.5}$. (C) and (D): Proportion of all simulation replicates where the location-location pair with the highest migration rate is correctly identified as such by $F_{ST}$ or $\hat{r}_{0.5}$, respectively. (E) and (F): Proportion of all simulation replicates the location-location pairs with the five highest migration rates are correctly classified as such as such by $F_{ST}$ or $\hat{r}_{0.5}$. **Fig T. Ranking and classification metrics for coalescent-simulated sequence data (subnational migration, model A)**. (A) and (B): Proportion of all location-location pairs that are ranked correctly by either $F_{ST}$ or $\hat{r}_{0.5}$, respectively, when compared to the migration rate specified in the coalescent simulation. Points show these values for 1000 independent simulation replicates over different numbers of individuals (*n*) and SNPs (*p*) used to calculate $F_{ST}$ or $\hat{r}_{0.5}$. (C) and (D): Proportion of all simulation replicates where the location-location pair with the highest migration rate is correctly identified as such by $F_{ST}$ or $\hat{r}_{0.5}$, respectively. (E) and (F): Proportion of all simulation replicates the location-location pairs with the five highest migration rates are correctly classified as such as such by $F_{ST}$ or $\hat{r}_{0.5}$.
(PDF)

## Acknowledgments

We thank Aimee R. Taylor for her comments on the manuscript.

## Author Contributions

**Conceptualization:** Tyler S. Brown, Olufunmilayo Arogbokun, Caroline O. Buckee, Hsiao-Han Chang.

**Data curation:** Tyler S. Brown, Olufunmilayo Arogbokun.

**Formal analysis:** Tyler S. Brown, Olufunmilayo Arogbokun, Hsiao-Han Chang.

**Investigation:** Tyler S. Brown.

**Methodology:** Tyler S. Brown, Caroline O. Buckee.

**Supervision:** Caroline O. Buckee, Hsiao-Han Chang.

**Visualization:** Tyler S. Brown.

**Writing – original draft:** Tyler S. Brown, Caroline O. Buckee, Hsiao-Han Chang.

**Writing – review & editing:** Tyler S. Brown, Olufunmilayo Arogbokun, Caroline O. Buckee, Hsiao-Han Chang.

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
