## [Decision Letter · Decision Letter 0]

9 Feb 2021

Dear Dr Brown,

Thank you very much for submitting your Research Article entitled 'Distinguishing gene flow between malaria parasite populations' to PLOS Genetics.

The manuscript was fully evaluated at the editorial level and by independent peer reviewers. The reviewers appreciated the attention to an important problem, but raised some substantial concerns about the current manuscript. Based on the reviews, we will not be able to accept this version of the manuscript, but we would be willing to review a revised version. We cannot, of course, promise publication at that time.

If you decide to revise the manuscript for further consideration at PLOS Genetics, please aim to resubmit within the next 60 days, unless it will take extra time to address the concerns of the reviewers, in which case we would appreciate an expected resubmission date by email to plosgenetics@plos.org.

[LINK]

We are sorry that we cannot be more positive about your manuscript at this stage. Please do not hesitate to contact us if you have any concerns or questions.

Yours sincerely,

Xavier Didelot

Associate Editor

PLOS Genetics

Hua Tang

Section Editor: Natural Variation

PLOS Genetics

Reviewer's Responses to Questions

**Comments to the Authors:**

Reviewer #1: The manuscript attempts to address statistical inference of a single use case of molecular(genomic) epidemiology in malaria control, that is, ranking gene flow between population pairs. Traditionally, this has been done with allele frequency based metrics such as Fst, however here the authors incorporate a more recently established measure of population connectivity, Identity by Descent (IBD) as measured by the parameter R here. Several key findings include that sample size, marker number and population levels and migration influence the ability to rank gene flow. The paper is written well though is a little technical with the mathematics for some general genetics readers, so some effort could be made to simplify some of the descriptions.

One point that stands out for me with IBD statistics is that they rely on high levels of relatedness (near clonality) to measure gene flow. However, these signals will be rapidly lost if a genotype migrated to a high transmission area. Thus, these approaches might be limited when comparing low to low transmission or high to low, but not high to high (as this paper suggests) or low to high transmission. This is something to consider and would be worth commenting on/considering for future analyses.

The real data set used was SNPs extracted from genomic data from clinical samples collected in multiple populations of the Greater Mekong Subregion. Here P. falciparum transmission is low and patchy, and finding positive cases is challenging through routine surveys, many thousands of individuals screened to find only a few samples. Therefore, passive collection of isolates from clinical cases may be the only feasible/affordable means of identifying isolates for genomic analysis. However clinical (febrile) cases, as the authors also point out in the discussion, may present a biased view of transmission dynamics, especially if substantial numbers of asymptomatic infections persist that are not being analysed. There are two issues here, inability to accurately define gene flow due to lack of sufficient samples numbers (also keeping in mind that low density infections are very difficult to genotype currently), and the fact that clinical cases may not present an accurate view of what is actually going on.

The use of the distance by road measures might not be accurate in all settings as in South East Asia, travel by motorbike, boat and foot might be alternative means. Thus a comparison to other distance measures might be warranted. Some of the co-authors have experience in spatial epidemiology so I suspect they have considered this. A comment on why the distance by road method was chosen would be appropriate.

Reviewer #2: Please find the attached review.

Reviewer #3: This is a well written manuscript describing approaches for measuring geneflow between populations, connectivity. It addresses an important question relevant to malaria elimination and the translation of genomic surveillance approaches for this deadly disease. It contrasted measures of geneflow based on allele frequencies (FST) and ancestry (measured by an IBD index, R). They employed real and simulated data, the latter controlling for migration rates and parameters that remain vastly unknown in natural parasite populations. Despite the importance of the findings, the manuscript is limited by a by the following:

1. The authors deliberately used data from the Greater Mekong region and only monoclonal isolates. This is convenient but far from the reality across malaria endemic regions. With the bulk of malaria transmission in Africa, there should be strong motivation to employ such approaches in this population especially for populations with reduced transmission, where monoclonality is high and elimination plans are being developed. Overall, how this will apply to more complex populations need further discussion.

2. As recognised by the authors, the bias in sampling is a major issue for connectivity studies based on genetic data. The authors could discuss the best sampling strategy for capturing connectivity at high confidence

3. The simulations allow for only 100 individuals per location without any justification for this number.

4. The filtered data retained 418 SNPs already biased for high minor allele frequencies. This will affect measurement of FST as low frequency variants are eliminated. Again, the choice on 0.35 MAF is based on a previous reference.

5. The index for pairwise relatedness is considered a summary for the genomes of the pairs. This will be true if the 418 SNPs are well distributed across all chromosomes. Except I missed this, data on how the retained loci span across the genome should be helpful.

6. Authors excluded sequences carrying the KEL1 haplotype. Connectivity studies as indicated in their introduction will help also to determine how antimalarial resistance flows between populations. Instead of excluding the isolates with KEL1 haplotypes, removal of SNPs around drug resistant gene sweeps could allow for tracking the flow of drug resistant isolates.

7. The Ne estimates used were from an African population in Senegal, obtained after years of interventions. Not sure they apply to SEA, and why this not determined. Determining Ne accurately is an issue as with other popgen parameters.

8. Phi was not defined and the value of 0.044 was used to scale the recombination rate. Only specialised readers will get this for an approach intended for future wider translation. How was this derived?

9. What is the added value of PRC, given R seems a robust measure of connectivity?

10. FST was not displayed on figure 1 to help readers

11. R is highly skewed. Low FST but high R will result from the presence of highly related pairs, which can be due to a common source outbreak rather than connectivity.

**Have all data underlying the figures and results presented in the manuscript been provided?**

Reviewer #1: Yes

Reviewer #2: Yes

Reviewer #3: Yes

PLOS authors have the option to publish the peer review history of their article (what does this mean?). If published, this will include your full peer review and any attached files.

Reviewer #1: No

Reviewer #2: **Yes: **Jack O'Brien

Reviewer #3: No

---

## [Decision Letter · Decision Letter 1]

12 Oct 2021

Dear Dr Brown,

We are pleased to inform you that your manuscript entitled "Distinguishing gene flow between malaria parasite populations" has been editorially accepted for publication in PLOS Genetics. Congratulations!

Yours sincerely,

Xavier Didelot

Associate Editor

PLOS Genetics

Hua Tang

Section Editor: Natural Variation

PLOS Genetics

Comments from the reviewers (if applicable):

Reviewer's Responses to Questions

**Comments to the Authors:**

Reviewer #2: Please find the attached review.

**Have all data underlying the figures and results presented in the manuscript been provided?**

Reviewer #2: Yes

PLOS authors have the option to publish the peer review history of their article (what does this mean?). If published, this will include your full peer review and any attached files.

Reviewer #2: **Yes: **Jack O'Brien

**Data Deposition**

http://datadryad.org/submit?journalID=pgenetics&manu=PGENETICS-D-20-01965R1

**Press Queries**

---

## [Editor Report · Acceptance letter]

24 Nov 2021

PGENETICS-D-20-01965R1 

Distinguishing gene flow between malaria parasite populations 

Dear Dr Brown, 

We are pleased to inform you that your manuscript entitled "Distinguishing gene flow between malaria parasite populations" has been formally accepted for publication in PLOS Genetics! Your manuscript is now with our production department and you will be notified of the publication date in due course.

With kind regards,

Agnes Pap

PLOS Genetics

On behalf of:
